# Analyzing the relationship between productivity and human communication in an organizational setting

**Arindam Dutta**[1]*, **Elena Steiner**[1], **Jeffrey Proulx**[2], **Visar Berisha**[1], **Daniel W. Bliss**[1], **Scott Poole**[2], **Steven Corman**[1]

**1** Arizona State University, Tempe, Arizona, United States of America, **2** University of Illinois at Urbana Champaign, Champaign, Illinois, United States of America

* adutta7@asu.edu

## Abstract

Though it is often taken as a truism that communication contributes to organizational productivity, there are surprisingly few empirical studies documenting a relationship between observable interaction and productivity. This is because comprehensive, direct observation of communication in organizational settings is notoriously difficult. In this paper, we report a method for extracting network and speech characteristics data from audio recordings of participants talking with each other in real time. We use this method to analyze communication and productivity data from seventy-nine employees working within a software engineering organization who had their speech recorded during working hours for a period of approximately 3 years. From the speech data, we infer when any two individuals are talking to each other and use this information to construct a communication graph for the organization for each week. We use the spectral and temporal characteristics of the produced speech and the structure of the resultant communication graphs to predict the productivity of the group, as measured by the number of lines of code produced. The results indicate that the most important speech and network features for predicting productivity include those that measure the number of unique people interacting within the organization, the frequency of interactions, and the topology of the communication network.

## Introduction

The "structural imperative" in network research [1] suggests that we can represent any organization as a network and look at the network as a determinant of behavior, culture, and the individuals within the organization. Organizational networks are generated and populated by human beings who are active agents with intentions, knowledge, and the ability to rationalize their actions. From interactions between individuals in an organization we can derive certain qualitative aspects like behavior, intentions, emotions and inter-employee relations of a workplace. These aspects play a large role in the effectiveness and productivity of an organization.

**Data Availability Statement:** Anonymized data is available from openICPSR: https://doi.org/10.3886/E130041V1. Non-anonymized data are not released because they could be used to identify individual

participants. Researchers can request access to the non-anonymized data by contacting Dr. Steven Corman (steve.corman@asu.edu), PI of this project, or the ASU IRB (Phone: 480-965-6788 | Fax: 480-965-7772 | Email: research.integrity@asu.edu).

**Funding:** Dr. Steven Corman NSF PD 11-8031 National Science Foundation https://www.nsf.gov/publications/pub_summ.jsp?ods_key=gpg15001&org=NSF The sponsors played no role in this manuscript.

**Competing interests:** The authors have declared that no competing interests exist.

In this paper we aim to directly study this relationship between productivity and communication, and report new methods for doing so.

While productivity is relatively straightforward to measure, existing studies measure communication indirectly, either through member self-reports of communication on rating scales [2, 3], through external raters' evaluation using global scales that assess communication behavior [3], as communication technology investment [4], or through questionnaires measuring more distal constructs such as communication satisfaction or perceived effectiveness [5, 6]. While these studies are useful, they can be challenged on the grounds that perceptions of communication do not correspond to actual communication behavior [7]. Direct observation is the "gold standard" for measuring communication and provides the most rigorous test of the communication-productivity relationship. Though several studies involving direct observation of communication behavior have been completed (for a review see [8]), these typically involved methods of human observation of small groups for short periods or unusual settings (for example Ham radio operators) where communication is routinely logged. Long-term studies based on objective observation are needed to supplement and validate current understanding of the relationship between communication and productivity.

Our general research question is:

What is the relationship between the amount of communication in an organization and its productivity? What are the factors that may moderate this relationship?

Several factors may moderate the productivity-communication relationship. One particularly important factor is the type of work the organizational unit in question does. For units engaged in the production of verbal outputs-such as plans, reports, audits and in those whose primary work involves interacting with clients or customers-such as those delivering education, therapy, or advice-an argument can be made that the greater the amount of communication, the higher the productivity. For units engaged in action or production, however, a different relationship would be expected: communication is good up to a point, but too much communication interferes with action or production. Moreover, in these units, high levels of communication may signal that they are experiencing difficulties and hence must engage in problem solving that requires high levels of communication. In this case, we can expect a non-linear relationship between communication and productivity, communication is positively related to productivity up to a point, past which it is negatively related. Since the organizational unit we are studying is engaged in producing software, we would expect an inverted-U shaped (2nd order polynomial) relationship between communication and productivity.

In this work, we estimate inter-employee communication networks in a software engineering organization using speech recordings. For a period of 3 years, all employees wore audio-recorders during their hours of work which recorded their conversations, and weekly communication graphs were estimated based on the detected speech. We use a simple speech activity detector, combined with inter-recorder correlations, to detect interactions between individuals and to construct daily communication graphs. In addition, we also measure several speech features that describe the speaking style of each individual. These features, which are defined in more detail in the S1 Appendix, include, pitch, temporal features (energy, zero crossing rate), spectral features (spectral centroid, spectral flux etc), and cepstral features (mel-scale frequency cepstral coefficients-MFCCs). Numerous studies have used these speech features to detect speakers and speech features such as emotions with high accuracy [9–17]. Each research has, in turn, linked various speech features to emotion. At the neurological level, emotions are known to have an impact on individual task performance [18, 19]. Emotion also influences individual behavior in task performance, citizenship and deviance [20]. Ashforth and Humphrey [21] reviewed the importance of emotion in organizational contexts, including its effects on motivation, leadership, and group dynamics. All of these have been associated with

performance in empirical research, for example, motivation, [22], leadership [23] and group dynamics [24]. It is important to study emotion alongside network structure because networks are a substrate of emotional contagion, and such contagion has been shown to influence group dynamics [25]. Therefore, we use a combination of networks and speech analysis to analyze the relationship between productivity and human communication in an organization. The method for this study was not intended to be applied by other organizations for practical purposes. Our immediate purpose in comparing productivity to detected interaction was to validate our detection method, i.e. to prove that the communication we detected has expected relationships to organizational outcomes. An additional purpose was to support a larger sponsored project, focused on discrepancies between observable and perceived communication [26].

## Method

### Organization setting and data collection

This study was approved by the Arizona State University IRB (Approval number: STUDY00003138), and written consent forms were obtained for participants. The setting for this research was the Software Factory (SF), a service unit at a large southwestern university providing software engineering services for funded research projects and university technology spinouts. SF had directors and work was led by a professional software engineer who managed student programmers using industry-standard engineering processes and were organized in forma, project-based teams. These characteristics put it squarely in the category of a professional organization [27]. It operated for 144 weeks from late 2002 to early 2005, and had 79 participants, including the manager, employees, clients, and researchers. Over this time, SF worked on 31 separate projects, developing applications for the social sciences, natural sciences, and education, and for internal use (such as an activity reporting system). The major steps of handling a project at the Software Factory consisted of four major processes:

- The business process,

- The development process,

- The design process, and

- The implementation process.

Typically, the initial business process involved the most senior people on the customer side (including the decision maker) and the highest-level SF personnel (one or more directors and a project manager). When the client had already identified one or more students to work on the project, they may also be in attendance. The development process included collaboration between the customer, project manager and the technical lead of the project. The major activities in this process involved validating with the customer, setting realistic customer expectations, and communicating to all SF personnel working on the project. The design process included the project manager, technical lead and the developers, and lastly the implementation stage involved the technical lead and the developers. These projects varied in terms of timescale and the number of SF personnel involved. Over the course of 144 weeks, there were instances where multiple projects existing at the same time, involving multiple employees, and some instances with an employee being involved in multiple projects at the same time. This study used only records from the 54 SF employees, because only employees made entries in a code repository and activity reporting system, data used in this paper.

The SF data is a unique dataset that aimed to accomplish, as nearly as possible, ubiquitous observation of a set of 79 employees and clients of the organization. The dataset contains

recorded audio data from participants between September-2002 and June-2005. Whenever they entered the dedicated SF facility, participants attached a digital recorder and lapel microphone, and logged in to a server which placed a time stamp on the recording. When leaving, they uploaded the recorded audio to a server for storage. The resultant dataset contains daily recordings of all SF employees and visitors (primarily clients) comprising approximately 7000 hours of time synchronized recordings. There was no evidence if employees ever chose to delete or not turn in recordings, it would have been reflected in our time-aligning analyses for cross-correlation mentioned in the later section. Also, people involved in SF said that after the first week or so, members tended to forget the recorders. The same has been reported in other studies doing long-term recording of participants. The participant recordings were created in digital speech standard (DSS) file formats, a compressed proprietary format optimized for speech. They were converted to an uncompressed WAV format using the Switch Sound File Converter software. The files were stored using a 6kHz sampling rate with 8-bits/sample.

In addition to the recordings, we analyzed the code written by employees at the SF. All codes were stored and managed using a Visual Source Safe (VSS) 6.0 repository. We used the VSS API to extract records from the repository. Each record included the filename, date, user, version, and changes, insertions, and deletions at check-in. From this information we were able to compute the number of lines of code at each check-in. In particular, we computed the total number of inserted, deleted and changed lines of code per employee per week. A total of 11276 entries of changes in LOC were recorded staring from the first week of March-2003.

The SF dataset affords a unique opportunity to obtain a holistic picture of work activity and communication in a small organizational unit over an extended period. In this analysis, we have used the audio recording from March-2003 to June-2005 (124 weeks), to build communication networks and extract speech features to predict the effective lines of codes obtained using VSS analysis.

Other studies in the literature have found that LOC is an effective measure of productivity in software organizations [28, 29].

## Approach

All analyses were done on a weekly basis. In case of communication graphs, individual interactions between any two individuals were detected using a simple cross-correlation scheme. Individual interactions were converted to a communication graph representing the frequency of interactions between any two individuals over the course of a week. From this graph, we extracted a set of features that describe the topology of the resultant network and denote that by, $G_w \in \mathbb{R}^{1 \times f_g}$, where $f_g$ is total number of graph features. In addition, we also extracted several speech features from the daily recordings and calculate two statistics (mean and variance) for these features across the whole week for all participants. These are defined as, $S_w \in \mathbb{R}^{1 \times (2 \times f_s)}$, where $f_s$ is total number of speech features. Thus, we had a total communication feature space defined by $C_w : (G_w \| S_w) \in \mathbb{R}^{1 \times (f_g + 2 \times f_s)}$ (where $\|$ is the concatenation operator).

We describe the details of how we estimate the communication graph and the feature extraction in the sections below. We then describe how we predict productivity using these features.

## Communication graph analysis

**Pair-wise communication detection.** To construct the communication graphs, we used cross channel signal analysis. The entire process of graph analysis can be subdivided into two main blocks, the construction of speech cross-correlation graphs and graph feature extraction as shown is Fig 1.

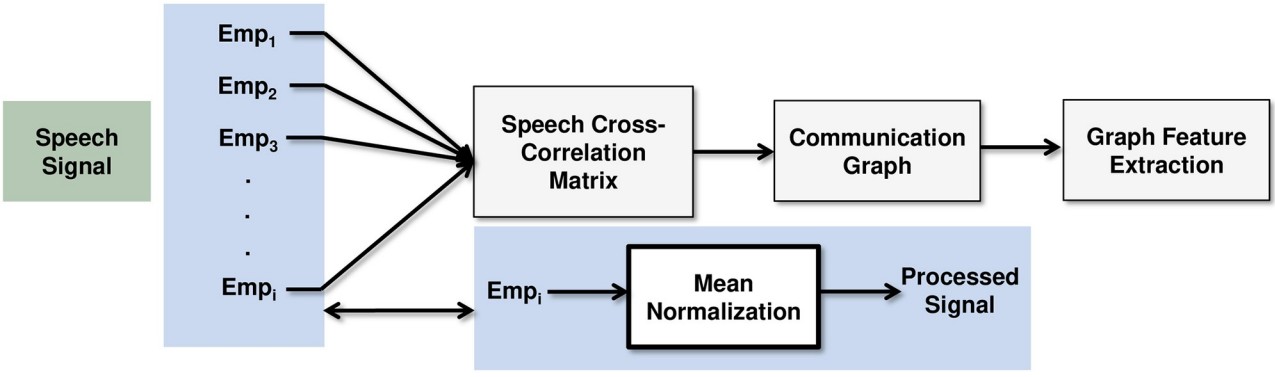

**Fig 1. Process chain for communication graph analysis.**

**Speech cross correlation graph.** As a pre-processing step we normalized the data by the mean to remove DC offset (caused by the analogue parts of the system that add a DC current to the audio signal), that causes significant interference with the audio signal, especially during signal processing. We investigated preliminary conversation detection performance on the SF data by using a two-stage approach. The first stage identified continuous segments of speech using an energy and spectral based detector; in the second stage, we use a pair-wise cross-correlation between one speaker's channel and the remaining channels to detect with whom that person was speaking. The basic idea behind this approach is that, if two individuals are speaking, their microphone will pick up each-other's speech and cross correlation will be high. A cross-correlation matrix was constructed using mean correlation weights between participant pairs across each day. The weights were calculated based the quantity of communication between participant pairs for an entire working day. The correlation matrix represents a proxy for the frequency of interactions between any two individuals. The same data can also be used to detect individual interactions and compare against manually coded data. Pairwise conversations between two speakers were detected by the algorithm and were presented to research assistants for manual coding. The daily cross-correlation matrices, which represent a proxy for frequency of interaction between two speakers, were averaged over the week to construct weighted communication graphs, with participants as nodes and the correlation weights as edges.

In the automated interaction detection, we used simple speech processing techniques from audio segments of both employees in a dyad to detect communication. First, we computed the short-time speech energy and spectral centroid (See S1 Appendix) for every 15 seconds frame and estimated thresholds to detect speech from the two features. Speech portions were detected using the two thresholds and non-speech portions were removed.

Next, we computed the covariance matrix between energy of speech segments from both microphones in a dyad. Two sets of thresholds were estimated based on the diagonal elements of the matrix, (a) $Th_1$, to determine if communication occurred (0 or 1, 2, 3) and (b) $Th_2$, to determine the direction of communication (1, 2 or 3).

**Validation of detection.** Before constructing the communication graphs based on pairwise cross-correlation, we validated the detections by comparing them to human coder classifications of the audio recordings as indicating network connections. We extracted 10 minute audio segments from a dyad from random working days. First we determined the total number of segments required to assess validity. Based on this we extracted that number of segments through random sampling from the audio corpus. External raters then coded the 15 second

segments regarding whether there was talk or silence in the segment and who was talking to whom. The specific classifications they could make were:

- Silence/noise (0)

- Employee 1 speaking (1)

- Employee 2 speaking (2)

- Both employees speaking (3)

We determined the minimum number of audio segments required to assess validity using the confidence interval equation,

$$N \geq \frac{\hat{p}(1 - \hat{p})}{\epsilon^2}$$

where $N$ is the minimum number of samples, $\hat{p}$ is the estimated population proportion and $\epsilon$ is the margin of error. With an error margin (variance per sample) of 5% and a $\hat{p}$ of 0.8, the minimum number of samples required is 64. In our analyses, a total of 75 ten minutes audio segments from random working days and between random dyads were used for communication validation. As Fig 3 indicates, there was 88% agreement between the coders and the automated detection (see next section for more details).

**Graph feature extraction.** After the graph was constructed using pairwise speech correlation, we extracted several topological features that aim to describe the nature of daily interactions. A total of 11 graph features were investigated in this work, which are described in more details in S1 Appendix.

*Basic graph descriptors.* We calculated the following basic graph descriptors:

- *Number of edges.* The total number of communication links present between employees in the network.

- *Number of nodes.* The total number of active employees present in the network.

- *Average degree.* Defined as the number of links that are incident on a particular employee. It is informative of total communication for individual employees.

- *Number of connected triples.* A count of the number of connected triples in the graph.

- *Number of cycles in a graph.* Defined as $m - n + c$, where $m$ is the number of links, $n$ is the number of employees and $c$ is the number of connected components. This indicates how connected the network is.

- *Graph energy.* The sum of the absolute values of the real components of the eigenvalues of the graph. They tell us about the structural complexity of the network. A structurally complex network has more differentiated interactions, which suggests members are working on different tasks in smaller groups and also that there is some interchange among these small groups.

*Graph centrality measures.* We computed the following graph centrality measures:

- *Degrees.* The average number of links adjacent to an employee node. This is an effective measure of the influence or importance of individual nodes on the network.

- *Average neighbor degree.* The average degree of adjacent or neighboring nodes for every vertex. We took the average of this measure across all nodes. This indicates the flow of communication around the organizational unit.

- *Eigen centrality*. The *i*-th component of the eigenvector of the adjacency matrix gives the centrality score of the *i*-th node of the network. The average eigen centrality across all nodes was computed for this study. This measure tells us about the quality of communication of an employee with others. This indicates the influence an employee over other employees in the organization.

  *Laplacian features*. We also calculated two Laplacian graph features.

- *Graph spectrum*. Defined as the eigenvalues of the Laplacian of the graph. This tells us about the frequency of communication in the organizational unit and its relationship to the nodes and link attributes.

- *Algebraic connectivity*. The magnitude of this value reflects how well connected the overall network is. It has been used in analyzing the robustness and synchronizability of networks.

  These features are estimated based on daily graphs. We average over the week to compute a weekly graph feature vector, $G_w \in \mathbb{R}^{1 \times 11}$, where 11 is total number of graph features investigated.

### Speech analysis

In addition to the graph features, we extracted speech features for every speaker from the data. These features carry information about speaker identity and various aspects of affect, which are important characteristics for predicting productivity.

**Speech feature extraction.** Speech features are extracted independently for every speaker (e.g. every recording channel). Prior to feature extraction, we remove the DC offset, and split the data into 1-second speech segments using hamming windows. All features are extracted at this scale.

A total of 35 different features were obtained from the audio data. Some of these pertained to whether there was a network linkage between actors and others pertained to properties of the linkages. In view of the exploratory nature of this research, we included the latter in order to capture a richer description of the nature of the links than a simple linked-not linked description would provide. As mentioned before, emotion affects productivity and these emotions can be recognized from variations in various aspects of speech. The speech features used for this study are mentioned below and described in details in S1 Appendix,

- *Pitch*. Features related to pitch contain information related to speaker emotions [9, 10, 13]. *Fundamental pitch frequency*, 12 *harmonics* and *harmonic ratio* were the pitch-related features that were investigated in this study.

- *Temporal features*. These features capture certain aspects of speaker emotion, like stress level, joy, excitement etc [9, 10]. We calculated the *zero-crossing rate*, *shot-time energy* and *energy entropy* from every one-second speech frame.

- *Spectral features*. These features carry the particulars of the frequency content of speech. They carry information about speaker identity and can help classifying a wide range of emotions [10, 11]. The spectral features investigated in this study are the *spectral centroid*, *spectral spread*, *spectral entropy*, *spectral flux* and *spectral rolloff*.

- *Cepstral features*. These features capture the characteristics of our auditory system based on changes in emotions, irrespective of language or gender. A significant number of speech emotion recognition (SER) research papers have identified these as one of the most efficient features for emotion classification [9–11, 13, 16]. Thirteen *Mel-frequency cepstrum coefficients (MFCC)* were extracted from 20 ms frames and averaged over 1 sec window.

We calculated the mean and variance of these features over the working days of a week to compute weekly speech feature vectors defined as, $S_w \in \mathbb{R}^{1 \times (35 \times 2)}$, where 35 is total number of graph features investigated and 2 is the number of statistics computed for each speech feature. Thus, together with the graph and speech features we had a combined communication feature set defined by, $C_w \in \mathbb{R}^{1 \times (11+70)} = \mathbb{R}^{1 \times 81}$.

## Measure of productivity

In this paper, the overall organization productivity, defined by the total lines of codes per week per employee ($LOC_w$) was used as the measure of productivity in the SF. The total LOC was calculated for each week as the sum of '*changed*', '*inserted*' and '*deleted*' LOC, as, $LOC_w =$ Changed + inserted + deleted LOC. The weekly LOC measures were converted to log scale to reduce the variable dynamic range. The average LOC per employee was calculated bu normalizing the LOC measure by the number of employees present during the particular week.

## Predicting productivity from communication

Regression methods allow us to summarize and study relationships between two continuous (quantitative) variables. One variable is regarded as the predictor, explanatory, or independent variable (in this case the weekly '*communication features*, $C_w$'), and the other variable, is regarded as the response, outcome, or dependent variable (in this case weekly '*productivity*, $LOC_w$'). We mentioned before that we should expect an inverted U-shaped relationship (polynomial of order 2) between communication and productivity. To apply this hypothesis, we first selected the communication features that exhibited such relationship. The selected communication features were then used to predict the organizational productivity. Since the variables are consecutive and evenly-spaced observations in time, it is a sequence of discrete-time data, where each data point is dependent on previously observed values. Consequently, We used a time-series regression model to predict productivity. In general, our regression model assumes productivity and the communication features are related to one another by

$$LOC_w = \mathcal{F}(C_w, t) + \epsilon$$

where $\mathcal{F}(C_w, t)$ is some mathematical operation (or model) showing productivity as a function of the input communication features and time (weeks), and $\epsilon$ is the prediction error. Fig 2 shows the block diagram of the prediction process and each block is described.

**Pre-whitening.**   Pre-whitening is required to remove autocorrelation and trends from the time-series variables, so that a meaningful relationship between the variables can be assessed. It concentrates the main variance in the data in a relatively small number of dimensions, and removes all first-order structure from the data. We implemented the ZCA whitening

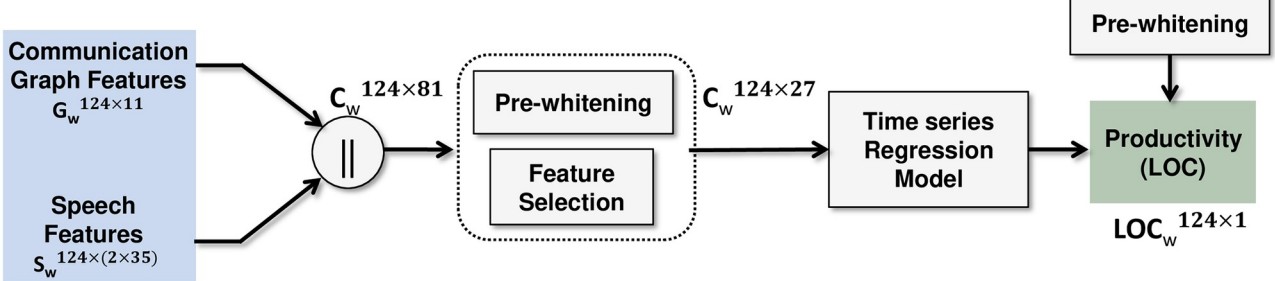

**Fig 2. Prediction process chain.**

transformation,

$$\hat{X} = \frac{X - \mu(X)}{\sqrt{cov(X)}}$$

where, $\mu(X)$ and $cov(X)$ are the mean and covariance matrix of time-series variable $X$. $\hat{X}$ is the transformed variable whose covariance matrix is the identity matrix. We pre-whitened all the independent variables ($C_w$) and the dependent variable ($LOC_w$).

**Feature selection.** We used a rank based feature selection method with a regression model ($\mathcal{F}(C_w)$) to evaluate correlation weights of each communication feature independently with 10-fold cross validation (in a 10-fold cross validation, the entire set is divided into 10 subsets, where 9 of them are used to train the regression model and one set for prediction). A support vector regression (SVR) model (see S1 Appendix) with a second order polynomial kernel (according to hypothesis) was used to find the association of each feature with the measure of productivity. Features with correlation weights above zero were selected for prediction analysis. Fig 2 shows that 27 communication features were selected from 81, which were given as input to the regression model.

**Time-series regression.** After selecting the most correlated features, they were used to predict productivity ($LOC_w$) using a time-series regression model. The SVR model with second order polynomial kernel was used as the base regression model. We can write the final model as

$$\mathcal{F}(C_w, t) = \sum_k \mathcal{F}(C_w(t - k))$$

To test the accuracy of the model $k$-steps ahead predictions were made at each data point, for $k = 0, 1, 2, 3, \ldots, 8$. Prediction for various time lags (1–8 weeks) were evaluated, to assess the dependency of productivity on past data.

## Results

### Pair-wise communication detection results

In the pair-wise communication detection, the four main classes were, "*Silence/noise*" (0), "*Employee 1 speaking*" (1), "*Employee 2 speaking*" (2), and "*Both employees speaking*" (3). The receiver operating characteristics (ROC) curve (see Fig 3) was used to illustrate the communication detection accuracy (0 or 1, 2, 3). The ROC curve was constructed by varying the threshold $Th_1$, and the optimum value of $Th_1$ was determined. Threshold $Th_2$ was determined after constructing confusion matrices for various $Th_2$ values. The threshold parameters for the best model were $Th_1 = 2.53e^{-5}$ and $Th_2 = 2.02e^{-5}$. We have shown the confusion matrix of the best detection model in Table 1.

Our method produced a good communication detection rate (AUC: 0.88), and on reviewing the results, we noticed that most of the false positives resulted because of the presence of other employees. We then constructed the daily communication graph using the above detection method, with correlation weights as edges connecting the employees present in the day. Thus in case of a communication scenario with more than two employees, the correlation weights will be high for any dyad with the speaker in it, while the correlation weights between other employees will be relatively low. For any focal individual the correlation weights between that individual will be high with anyone they address, while those between other speakers who might be detected in the background is lower.

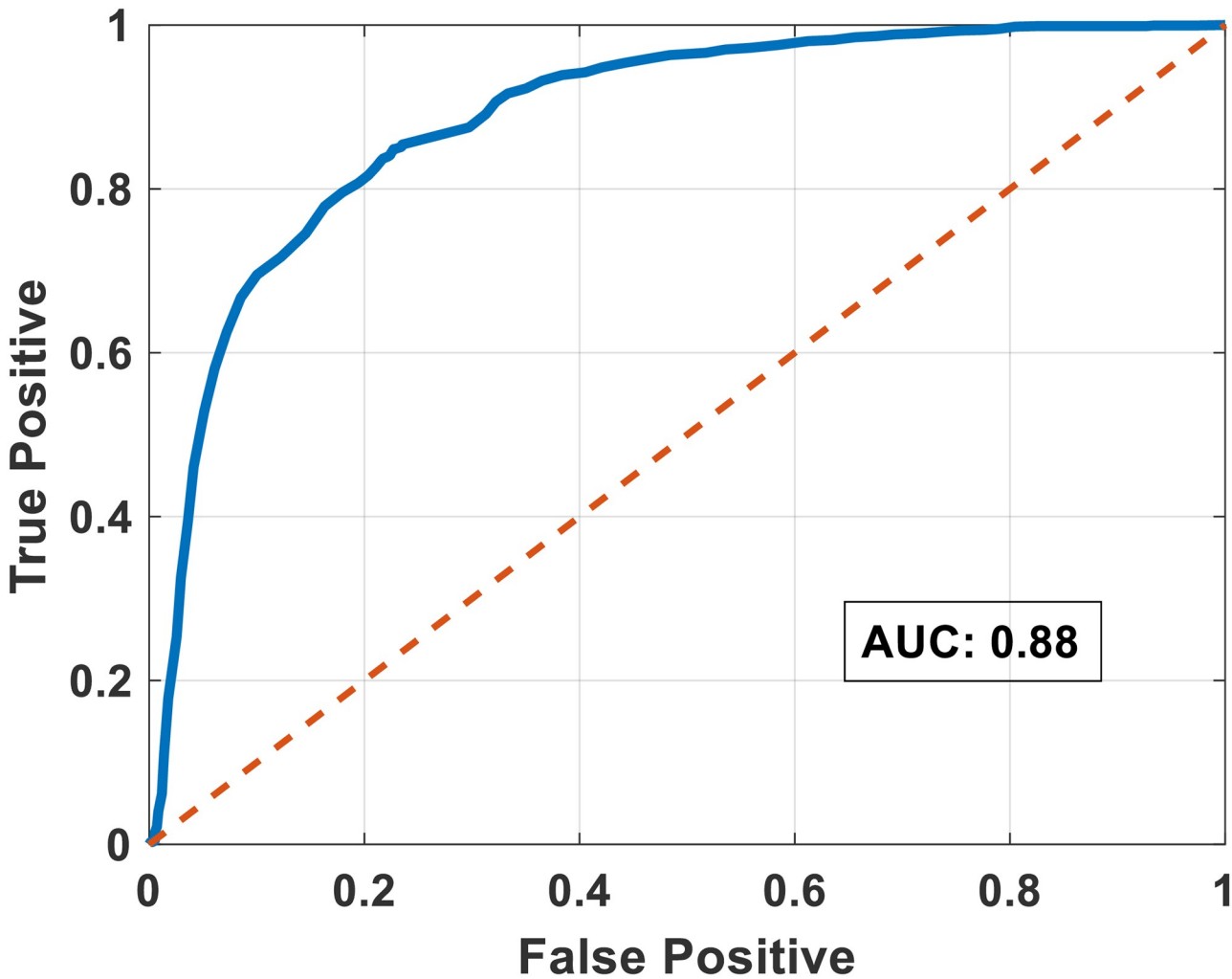

**Fig 3. Receiver operating characteristics curve for communication detection; area under curve (AUC) = 0.88.**

### Predicting productivity from communication

**Feature selection.** We computed the correlation weights for each communication feature while predicting productivity. Fig 4 shows the average merit of the features based on correlation weights achieved while predicting $LOC_w$. It can be seen that almost all the graph features (10 out of 11) had positive correlation weights. Among the weekly speech features, the MFCC coefficients (1, 2, 3, 4, 5, 6, 8), the spectral and energy entropy (mean), fundamental frequency

**Table 1. Confusion matrix for the best detection model; each element is shown in terms of number of 15 seconds segments.**

|  | | Coder | | | |
| --- | --- | --- | --- | --- | --- |
|  | **Class** | **0** | **1** | **2** | **3** |
| Tool | 0 | **1390** | 183 | 227 | 74 |
|  | 1 | 51 | **201** | 32 | 105 |
|  | 2 | 70 | 14 | **309** | 98 |
|  | 3 | 16 | 42 | 50 | **138** |

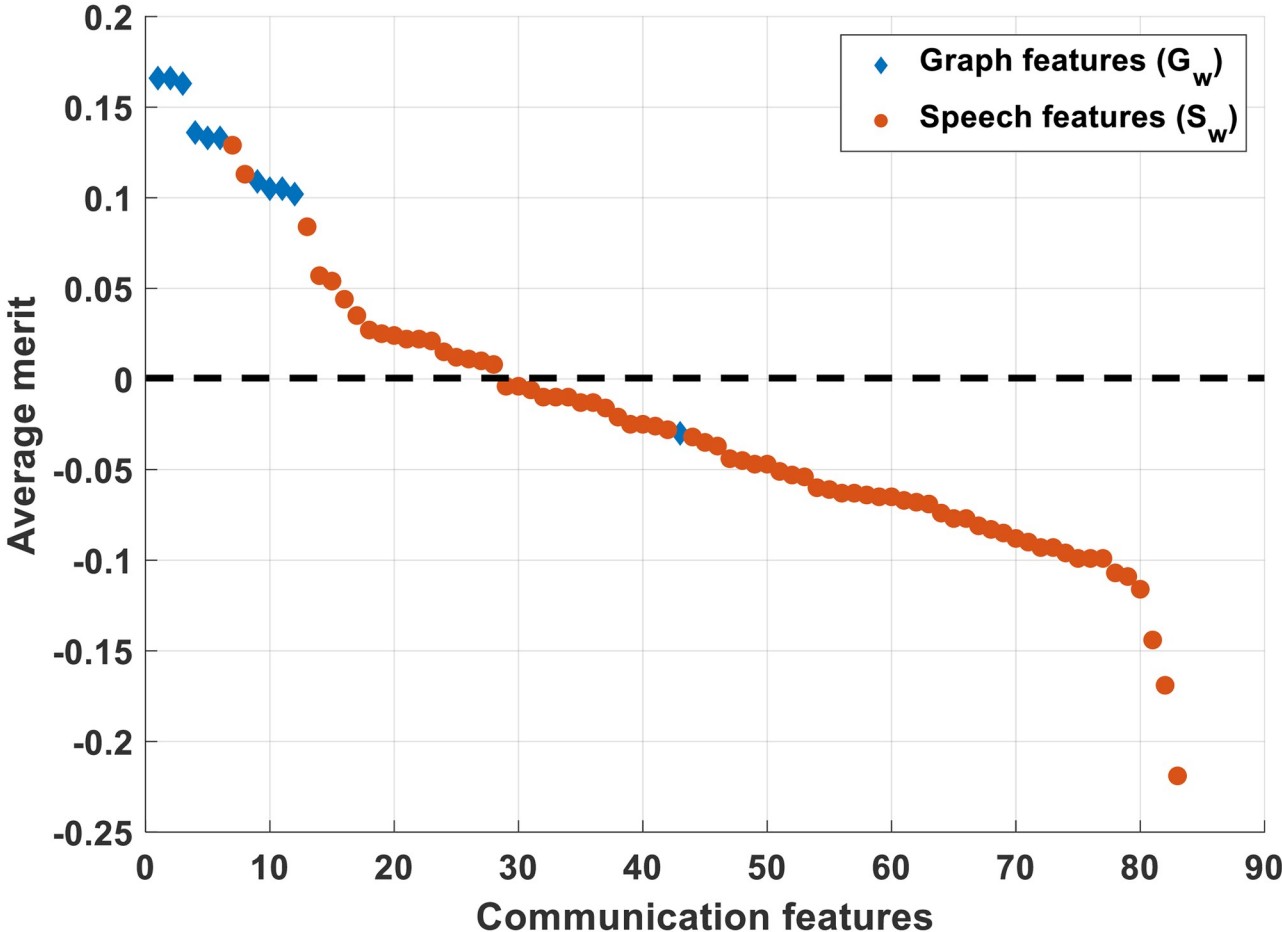

**Fig 4. Correlation coefficients for each selected communication feature.**

(variance), spectral roll-off (mean) and spectral centroid and spread (mean) were positively correlated. Comparing the two types of communication features, the graph features had higher correlation weight than the speech features. The number of nodes, average neighbor degree, algebraic connectivity, graph energy and graph spectrum were the features with highest average merit.

**Time-series prediction of productivity.** To analyze the communication-productivity relationship we made $k$-steps time-series prediction of $LOC_w$ at each data point using the selected communication features. We used lags of upto six weeks to analyze how much the productivity depend on previous weeks' communication. The mean absolute error (MAE), mean absolute percentage error (MAPE), root mean squared error (RMSE) and direction accuracy (DA) were measured to evaluate the accuracy of the time-series model. The time-series model implementation was done in WEKA 3.8 [30]. Fig 5 shows the $k$-steps ($k$ = 1, 2, 4, 8) prediction result using a lag of one week. The accuracy parameters are shown in Table 2 for 1 week and 6 weeks lags. Fig 6 shows the MAPE for different lags (1 to 8 weeks).

It can be seen that, using 1-week previous information, we can predict productivity ($LOC_w$) with an error of 7.2–9.8% (1–8 steps ahead prediction). This is error is reduced to 2.2–5.6%, when we use information from the previous 6 weeks. The direction accuracy also improves from 71–77% to 83–92%.

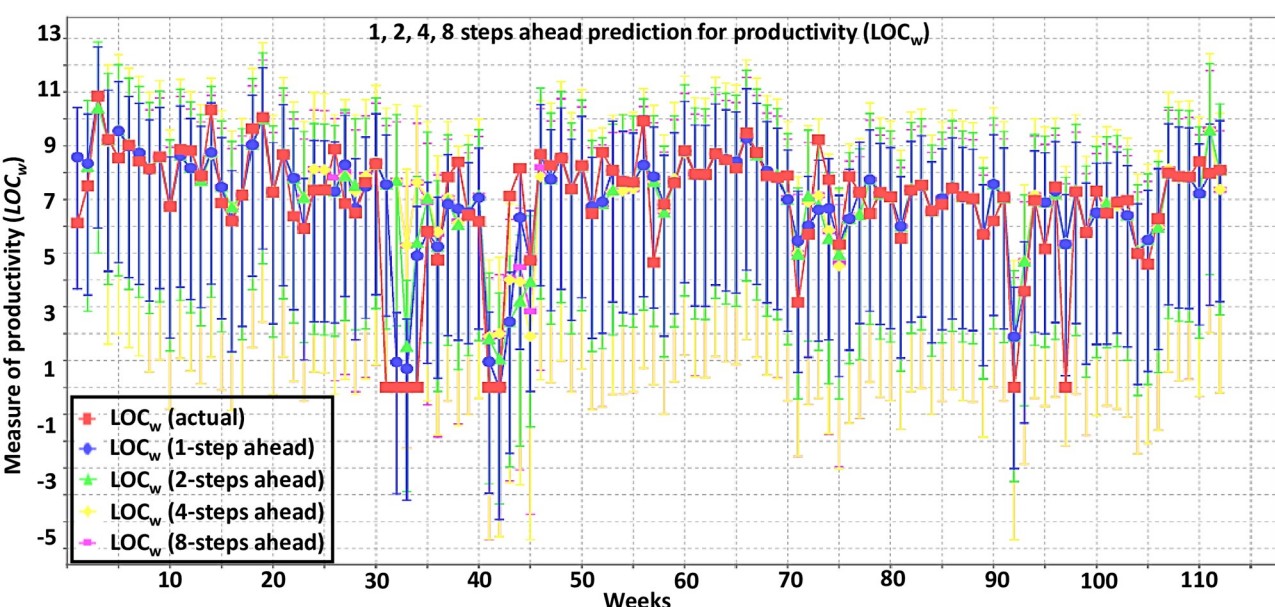

**Fig 5. 1, 2, 4, 8 steps ahead prediction for productivity ($LOC_w$) using 1-week past information.**

## Discussion

From the results we can conclude that communication is strongly related to productivity in an organization. Table 2 suggests that we can predict organizational productivity with high accuracy with mean absolute error less than 10%. We hypothesized before, that communication and productivity share a non-linear relationship (polynomial of order 2), and we made use of that relationship in the regression model. With the use of a second order polynomial kernel SVR model, we selected the communication features and used to same model to do a time-series forecasting of productivity. The results are also suggestive of the fact that the prediction accuracy improved as we used more previous information. Though comparisons are difficult due to differences in methods and measures, this study shows a stronger correlation between communication and performance than previous research. In [6], the authors found a relationship of $r = 0.27$ between two-way interaction and effectiveness. In [31], only a small $r = 0.02$ correlation between communication satisfaction and productivity was reported. It is possible

**Table 2. Accuracy of time-series model used to predict productivity ($LOC_w$) using communication features ($C_w$).**

|  |  | MAE | RMSE | MAPE % | DA % |
|---|---|---|---|---|---|
| 1 week lag | 1-step | 0.64 | 1.36 | 7.2 | 77.5 |
|  | 2-steps | 0.83 | 9.18 | 9.2 | 74.5 |
|  | 4-steps | 0.96 | 1.80 | 9.9 | 71.3 |
|  | 8-steps | 0.97 | 1.79 | 9.8 | 73.1 |
| 6 weeks lag | 1-step | 0.17 | 0.58 | 2.2 | 92.5 |
|  | 2-steps | 0.25 | 0.75 | 2.8 | 89.5 |
|  | 4-steps | 0.34 | 0.83 | 3.2 | 89.4 |
|  | 8-steps | 0.51 | 1.07 | 5.9 | 83.8 |

MAE: Mean absolute error; RMSE: Root mean square error; MAPE: Mean absolute percentage error; DA: Direction accuracy.

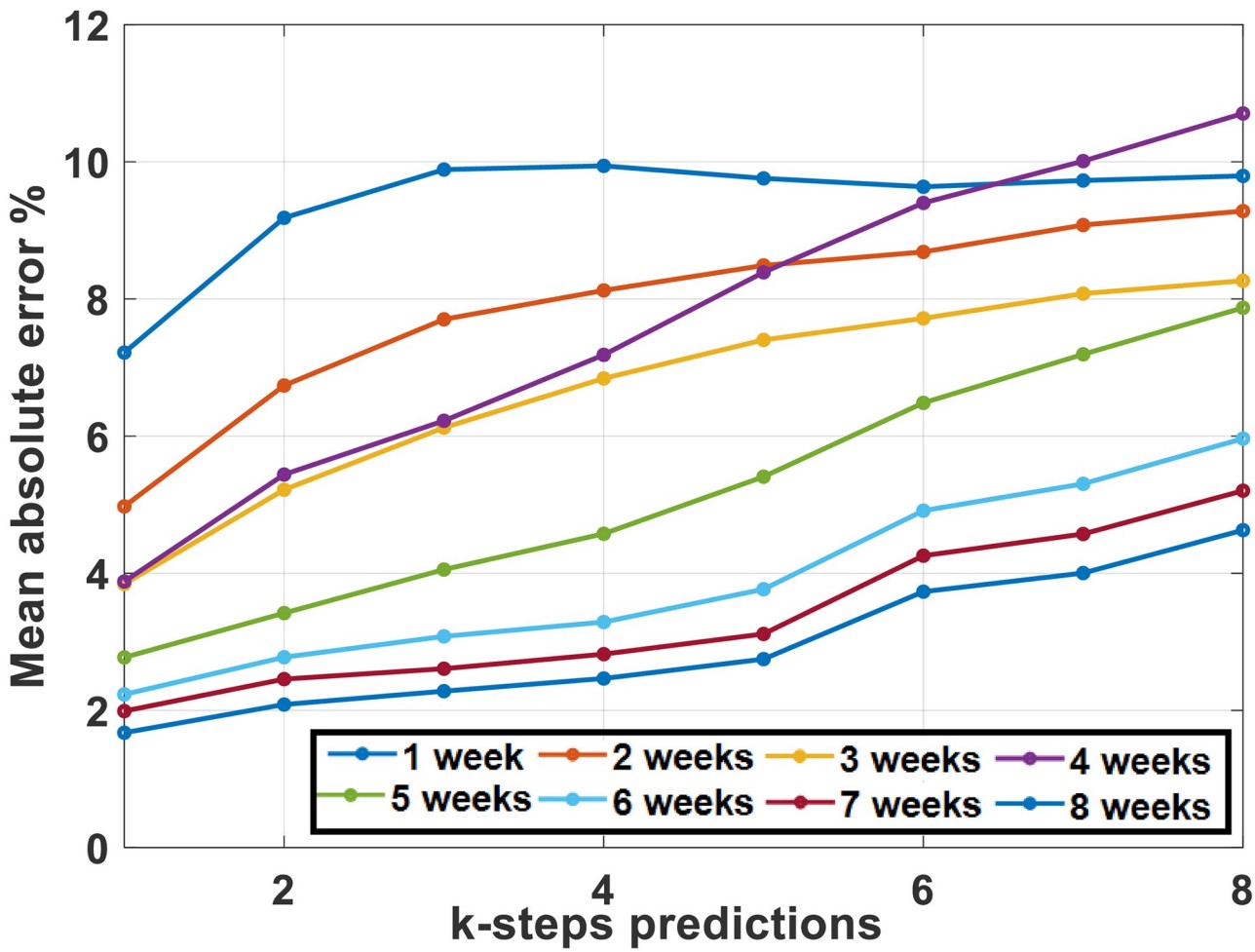

**Fig 6. Mean absolute percentage error while predicting productivity ($LOC_w$) for different lags (1–8 weeks).**

that the more long-term, detailed, objective measurement of both communication and productivity in this study allowed the relationship between the two variables which to most is common sense to be more accurately estimated.

The results from Fig 4 indicate the communication graph features played a more important role than speech features in predicting the dependent variables. Among the top graph features, algebraic connectivity, number of nodes and average neighbor degree signify the total number of employees and frequency of interactions between them and graph energy and graph spectrum tells us about the structural complexity of the network. From the speech features, the mean MFCC coefficients are likely tapping into the number of speakers in the graph; the spectral and the energy variability features are likely measuring the number of speakers and frequency of interactions. It is interesting that the fundamental frequency variability is a measure of productivity. This could be a proxy for gender diversity in the organizational unit, although this most certainly requires additional study.

It is important to note that while this study reveals some relationship between communication and productivity, it does not mean that this relationship is causal. It is unknowable from out data whether it is the productivity that induces a change in the network or whether the network induces a change in productivity.

The method described in this paper makes it possible to convert audio-recordings among members of an organization into communication network measures. As such it should be useful to group researchers, who often record all members of a group, and to those organizational researchers who record an entire unit or organization. While the data requirements for the method are demanding, it yields a much more accurate and potentially more valid measure of communication networks than do currently utilized questionnaire methods.

The best choice of a productivity measure can be argued here. Both changed and inserted lines of codes are important measures that cannot be neglected, when it comes to programmer productivity. The inclusion of deleted lines of codes is debatable, as those can be errors or bugs in previously-written codes, that can said to be counter-productive. But at the same time, it can argued that deletion mean shortening of code or making it more compact using improved logic, which is an important aspect of productivity.

This study is unique in terms of organizational communication as it involves long-term, objective, quantitative analysis showing the relationship between a human communication network and productivity in an organization. We have used speech recordings from employees in a software organization to estimate communication networks and extract speech features over a period of 3 years. Effective lines of code was used as the measure of productivity which we attempted to predict using both communication network and speech features. It was found that there exists a moderate relationship between communication and productivity in an organization and it depends on the number of employees, the frequency of conversation between them and the topography of the network. Further investigation can be done by including other forms of communication like, email, texts etc. Besides that, more complex graphs with multiple modules (employee, project, task) can be investigated, which can be a better representative of an organizational setting model. Although, project deadlines were not a prominent feature of SF work because it used extreme programming (XP) as its software development process, it could be interesting to study the communication productivity relationship for different project types and deadline situations. This study does not capture how the communication quantity or speech patterns are affected by specific job stages of a project and how the job stages drive the overall productivity. Since multiple projects overlapped over the whole timeline with employees working on multiple projects at the same time, analyzing various job stages remains a limitation of this study. It requires a more precise analysis of the communication pattern and productivity at various job stages in a project and compare the relationship across various job stages. Furthermore, we can also investigate on productivity on a personal level by analyzing the relationship between communication and productivity for individual employees in the organization.

## Supporting information

**S1 Appendix. Speech signal features.**
(PDF)

## Author Contributions

**Conceptualization:** Arindam Dutta, Visar Berisha, Daniel W. Bliss, Scott Poole, Steven Corman.

**Data curation:** Elena Steiner, Jeffrey Proulx, Steven Corman.

**Formal analysis:** Arindam Dutta, Visar Berisha.

**Funding acquisition:** Scott Poole, Steven Corman.

**Investigation:** Elena Steiner, Jeffrey Proulx, Visar Berisha, Daniel W. Bliss, Scott Poole, Steven Corman.

**Methodology:** Arindam Dutta, Visar Berisha, Daniel W. Bliss, Scott Poole.

**Project administration:** Scott Poole, Steven Corman.

**Resources:** Elena Steiner, Jeffrey Proulx, Scott Poole, Steven Corman.

**Software:** Arindam Dutta.

**Supervision:** Visar Berisha, Daniel W. Bliss, Scott Poole, Steven Corman.

**Validation:** Elena Steiner, Jeffrey Proulx, Visar Berisha, Steven Corman.

**Visualization:** Arindam Dutta.

**Writing – original draft:** Arindam Dutta, Visar Berisha.

**Writing – review & editing:** Arindam Dutta, Elena Steiner, Visar Berisha, Daniel W. Bliss, Scott Poole, Steven Corman.

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
