## [Decision Letter · Decision Letter 0]

15 Feb 2021

PONE-D-20-41110

Analyzing the relationship between productivity and human communication in an organizational setting

PLOS ONE

Dear Dr. Arindam Dutta,

Thank you for submitting your manuscript to PLOS ONE. After careful consideration, we feel that it has merit but does not fully meet PLOS ONE’s publication criteria as it currently stands. Therefore, we invite you to submit a revised version of the manuscript that addresses the points raised during the review process.

Read the reviewer comments carefully and provide detailed answer for every query individually and also incorporate required changes in the manuscript.

We look forward to receiving your revised manuscript.

Kind regards,

Nersisson Ruban, Ph.D

Academic Editor

PLOS ONE

Journal Requirements:

Reviewers' comments:

Reviewer's Responses to Questions

**Comments to the Author**

1. Is the manuscript technically sound, and do the data support the conclusions?

Reviewer #1: Partly

Reviewer #2: Yes

2. Has the statistical analysis been performed appropriately and rigorously? 

Reviewer #1: I Don't Know

Reviewer #2: Yes

3. Have the authors made all data underlying the findings in their manuscript fully available?

Reviewer #1: No

Reviewer #2: Yes

4. Is the manuscript presented in an intelligible fashion and written in standard English?

Reviewer #1: Yes

Reviewer #2: Yes

5. Review Comments to the Author

Reviewer #1: This paper uses very detailed voice recording data from what I think is a kind of student software consulting organization at a University over the course of almost 3 years to study whether measures of voice communication can predict productivity. The voice data used in the study is very unique in how complete it is and how long a time period it covers and it allows the authors to generate measures of communication between pairs of individuals inside the organization (employees and clients). The authors find that speech and communication network characteristics can predict software programmer productivity.

I do not have a sufficient understanding of voice measures to comment on the validity what seems to be heroic work by the authors to generate useful measures of communication networks and speech characteristics.

While I think a better understanding of how communication relates to productivity in organizations is a very important question, I am not convinced this study advances our understanding of this question. There are several reasons for this, which I separate out below.

1. The authors do not provide sufficient information about the organization they are studying, except that the work being performed by employees is software programming. The authors argue that the type of work being performed should affect the relationship between communication and productivity, which I agree with, but so should, for instance, the organizational structure, and the sales or work cycles. From what I can gather on my own, it seems as though the software factory is a student run software consultancy which suggests a very flat organizational structure in which most conversations may be between “equals”. It also suggests that work occurs when there is a project available, and otherwise no work occurs. Without understanding how the organization functions and how often work occurs in the organization, it is difficult to know the extent to which the findings generalize to other organizations.

Moreover, the authors mention that their voice data includes conversations between programmers and between programmers and clients. It seems to me like the type of communication that is desirable differs depending on whether a programmer is speaking with a fellow programmer or with a client. I may have missed a discussion of whether the authors drop the client-facing conversations, but I was surprised this wasn’t emphasized more. Given that the measure of productivity used in the paper does not directly capture how satisfied a client was or how big a contract the programmers got, I don’t think the conversations with clients should be included in the study.

2. As the authors point out, the study is not able to determine whether different types of communication patterns cause more or less productivity, but rather whether they can be useful to predict how well the organization is doing (in this case, how productive workers will be). For instance, communication frequency may be a proxy for how excited or engaged workers are in the job or how frustrated they are with the client’s demands. If speech and communication network characteristics are easier to observe and measure than other things like employee attitudes, that using them to predict productivity seems like it could be a very useful tool for employers to improve performance by helping them catch potential problems before they become disasters.

However, I do not think the authors did enough to explain how feasible or practical it is for employers to adopt this in practice. I imagine many employees would object to being recorded while working, and it may be hard to retain high quality employees if they were required to record themselves while at work. Given the importance of communication frequency for predicting productivity, partial recordings may not be sufficient. Generating useful data out of the voice recordings would also be expensive time-wise. Thus, it is unclear to me what the practical takeaways of the paper are.

3. I don’t think my previous comment would be important if the study improved our understanding of why people communicate the way they do when they are more or less productive. As I mention above, there are many reasons why communication patterns may change at the same time as productivity changes but the authors do not tell us anything about these potential mechanisms. I think the authors could do more with their data in this regard. For instance, perhaps they could get data on when project deadlines are and test whether the relationship between communication out productivity changes when workers are facing more urgency. Similarly, perhaps the authors could get information on when students are working on different types of jobs (e.g. maybe students participated in hack-a-thons at some points that were not contracted for). These types of tests may move us forward in our understanding of how speech and communication patterns differ depending on the circumstances, and how those differences relate to productivity.

4. This is a more minor comment, but I wondered if employees would choose to delete or not turn in recordings of less appropriate discussions, including disagreements. If this is possible, perhaps this might explain the lack of findings on speech signal features? Some discussion of possible measurement error of these features might be helpful for understanding how much we can take away from the findings.

5. I am also a bit concerned about mismeasurement of productivity in the lines of code changed may not capture code quality. The authors acknowledge that deletions may or may not improve code quality, but on the converse, additions may or may not improve code quality. At the extreme, if one programmer is particularly frustrated with a co-worker or client, they may write a number of confusing lines that could make it hard for subsequent programmers to successfully edit the code. There are some relatively standard measures of code quality that the authors could have external programmers evaluate the organization’s code on (e.g. complexity, coupling).

Reviewer #2: I think this is great. I like the analyses to measure communication among this group of employees. The sample size is a bit small for my taste, but overall it's quite sound. I think this could be an important paper that change the way we measure communication in the future.

6. PLOS authors have the option to publish the peer review history of their article (what does this mean?). If published, this will include your full peer review and any attached files.

Reviewer #1: No

Reviewer #2: No

---

## [Author Response · Author response to Decision Letter 0]

4 Mar 2021

We would like to sincerely thank the reviewers for their valuable suggestions. All the suggestions were very helpful, and we have tried our best to work on those suggestions and made necessary changes. The response to all the comments made by the reviewers have been addressed in the 'Response to Reviewers' letter, which also reflects the changes made in the paper.

---

## [Decision Letter · Decision Letter 1]

17 Mar 2021

PONE-D-20-41110R1

Analyzing the relationship between productivity and human communication in an organizational setting

PLOS ONE

Dear Dr. Dutta,

Thank you for submitting your manuscript to PLOS ONE. After careful consideration, we feel that it has merit but does not fully meet PLOS ONE’s publication criteria as it currently stands. Therefore, we invite you to submit a revised version of the manuscript that addresses the points raised during the review process.

Dear author, please address the concern raised by reviewer-1.

We look forward to receiving your revised manuscript.

Kind regards,

Nersisson Ruban, Ph.D

Academic Editor

PLOS ONE

Journal Requirements:

Reviewers' comments:

Reviewer's Responses to Questions

**Comments to the Author**

1. If the authors have adequately addressed your comments raised in a previous round of review and you feel that this manuscript is now acceptable for publication, you may indicate that here to bypass the “Comments to the Author” section, enter your conflict of interest statement in the “Confidential to Editor” section, and submit your "Accept" recommendation.

Reviewer #1: All comments have been addressed

Reviewer #2: All comments have been addressed

2. Is the manuscript technically sound, and do the data support the conclusions?

Reviewer #1: Partly

Reviewer #2: Yes

3. Has the statistical analysis been performed appropriately and rigorously? 

Reviewer #1: I Don't Know

Reviewer #2: Yes

4. Have the authors made all data underlying the findings in their manuscript fully available?

Reviewer #1: No

Reviewer #2: No

5. Is the manuscript presented in an intelligible fashion and written in standard English?

Reviewer #1: Yes

Reviewer #2: Yes

6. Review Comments to the Author

Reviewer #1: Thank you very much for your response to my comments. I appreciate the additional details on the work setting, and the inclusion of statements about the extent to which employees could have deleted their conversations and about the methods not being intended for adoption by organizations.

However, I don't think you sufficiently addressed my concern about speech patterns and productivity being both affected by some additional variable (I suggested deadlines, but it could also be the different stages of the task as outlined by the authors in the revised draft). Thus, it may look like certain speech patterns contribute to productivity in the data but in reality, it could be that certain job stages drive certain speech patterns and those same job stages generate different amounts of coding. I think this is an important limitation of the current paper, and at a minimum, should be mentioned in the conclusion as such.

Reviewer #2: As I said before, I think this is an interesting study that could have important contributions to the field. I think it should be published.

7. PLOS authors have the option to publish the peer review history of their article (what does this mean?). If published, this will include your full peer review and any attached files.

Reviewer #1: No

Reviewer #2: No

---

## [Author Response · Author response to Decision Letter 1]

26 Mar 2021

We would like to sincerely thank the reviewers for their valuable suggestions. All the suggestions were very helpful, and we have tried our best to work on those suggestions and made necessary changes. The response to all the comments made by the reviewers have been addressed below, which also reflects the changes made in the paper.

---

## [Editor Report · Decision Letter 2]

5 Apr 2021

Analyzing the relationship between productivity and human communication in an organizational setting

PONE-D-20-41110R2

Dear Dr. Dutta,

We’re pleased to inform you that your manuscript has been judged scientifically suitable for publication and will be formally accepted for publication once it meets all outstanding technical requirements.

Kind regards,

Nersisson Ruban, Ph.D

Academic Editor

PLOS ONE

---

## [Editor Report · Acceptance letter]

21 Jun 2021

PONE-D-20-41110R2 

Analyzing the relationship between productivity and human communication in an organizational setting 

Dear Dr. Dutta:

I'm pleased to inform you that your manuscript has been deemed suitable for publication in PLOS ONE. Congratulations! Your manuscript is now with our production department. 

Kind regards, 

on behalf of

Dr. Nersisson Ruban 

Academic Editor

PLOS ONE